# Exploring the Sensitivity Range of Underlying Surface Factors for Waterlogging Control

Yang Liu [1], Xiaotian Qi [2], Yingxia Wei [2] and Mingna Wang [1,*]

1    School of Civil Engineering, Tianjin University, Tianjin 300072, China; l.y0719@163.com
2    School of Environmental Science and Engineering, Tianjin University, Tianjin 300072, China;
     qixiaotian1997@163.com (X.Q.); weiyingxia7@163.com (Y.W.)
*    Correspondence: mingna.wang@tju.edu.cn

**Abstract:** To mitigate the incidence of waterlogging to livelihoods and property security, a combination of management measures has been necessary to achieve optimal benefits, reducing the risk caused by waterlogging to the development of the urban ecology. Thus, this study aims to analyze the sensitivity and sensitivity range of management measures under different rainfall conditions, focusing on establishing a foundation for their combined implementation. Based on different rainfall scenarios, the modified Morris method is employed to assess the sensitivity of key factors and subsequently determine their respective sensitivity ranges. The findings reveal that the sensitivity rankings for total overflow volume and maximum pipe flow are as follows: pipe volume per hectare (PV-H), proportion of impervious area (P-Imperv), and slope. Additionally, analyzing the variation pattern of sensitivity with factors highlight the high sensitivity ranges. As for total overflow volume, a very high sensitivity is observed when the P-Imperv ranges from 36.8% to 82.7% (Niujiaolong community) and from 82.7% to 94.5% (Zhuyuan community). Similarly, when PV-H is less than 148 (Niujiaolong community) and 89.6 (Zhuyuan community), the sensitivity of PV-H to total overflow volume is very high. Nevertheless, the slope had a lower influence on the sensitivity in the study areas. These findings provide a complete analysis of the management measures sensitivity, which can be valuable for creating optimal urban waterlogging management systems.

**Keywords:** waterlogging management; modified Morris method; sensitivity analysis; sensitivity range





## 1. Introduction

The increase in city expansion and the development of flood-prone areas have created an elevated number of regions that are vulnerable to waterlogging [1]. Moreover, frequent occurrences of heavy rainfall have further exacerbated the incidence of waterlogging events, e.g., from 2008 to 2018, a large amount of waterlogging incidents were reported annually in over 150 cities across China. In 2018, urban waterlogging affected 55.77 million people, generating CNY 16.15 billion (USD 2.347 billion) in economic losses [2]. Not confined to China, waterlogging constitutes a threat in various countries worldwide [3]. As one of the most severe urban disasters, waterlogging significantly impedes urban development [4].

In order to manage and mitigate urban waterlogging, two primary approaches are commonly employed. Firstly, significant emphasis is placed on the renovation of drainage systems, encompassing measures such as enlarging the pipe diameter of drainage networks, constructing additional drainage networks, and dredging watercourses. Various studies have proposed diverse renovation schemes for drainage networks, including those based on waterlogging risks [5]. Through the construction of drainage network models and the calculation of overflood depth and duration across different return periods, high-risk areas of urban waterlogging can be identified, facilitating targeted system renovation. Proposing new reconstruction methods based on the concept of urban drainage system resilience is

also a focus of research [6]. The reconstruction of drainage systems must adhere to peak flow discharge requirements. With the expansion of cities, the proportion of impervious areas is increasing, thereby simultaneously amplifying runoff volume and reducing time of concentration, consequently promoting elevated peak flow. To address these issues, it becomes imperative to implement source control measures for rainfall runoff. The concept of low impact development (LID) was initially proposed in the United States with the objective of managing stormwater runoff and mitigating pollution through decentralized and small-scale source control measures [7]. China has considered local conditions and introduced the sponge city concept, which aims to achieve rainwater infiltration, stagnation, storage, purification, utilization, and discharge [8,9]. Comparably, other countries have proposed comparable concepts, such as sustainable urban drainage systems (SUDS) in the United Kingdom and water-sensitive urban design (WSUD) in Australia. These concepts aim to address surface runoff by reconstructing the underlying surface, and effectively control waterlogging. Common reconstruction measures include the implementation of rain gardens, permeable pavements, and grass swales [10–15]. The reconstruction of underlying surfaces plays a crucial role in effectively managing urban waterlogging [4].

To optimize the management measures of waterlogging control, it is vital to prioritize factors that demonstrate sensitivity to waterlogging when selecting appropriate reconstruct measures. Several studies have examined the correlation between waterlogging events and underlying surface factors to evaluate their impact on waterlogging [16,17]. However, it should be noted that the underlying surface is subject to constant change, which implies that the underlying surface during a waterlogging event may differ from the obtained underlying surface information. In particular, existing research has focused on land-use, slope, and drainage systems as the primary sensitive factors influencing urban waterlogging [18]. Nonetheless, the sensitivity of these factors may vary with changes in their characteristic values. For instance, as the P-Imperv expands, the impact of land-use on runoff may emerge as the dominant factor [19]. Current studies on reconstruction factor selection tend to be rudimentary and often explore combined reconstruction schemes using methods of scenario analysis to determine the optimal combination [20]. However, under diverse rainfall conditions, the factors sensitive to urban waterlogging may undergo changes, thereby impacting the rationale behind reconstruct measures. Analyzing the sensitivity of factors under different conditions enables a more precise assessment of suitable retrofit measures. Moreover, alterations to underlying surface factors should remain within a reasonable range. Beyond a certain threshold, modifying these factors becomes impractical as they become less sensitive, rendering such changes inconsequential.

The impact of underlying surfaces on waterlogging is often investigated using both qualitative and quantitative approaches. Qualitative studies analyze topographical features, urban development, and urban drainage systems within the current urban context to examine the inducements of urban waterlogging [21]. Correlation analysis is commonly employed in qualitative studies to explore the relationship between underlying surface factors and waterlogging, thereby identifying primary influencing factors [16,17]. The research predominantly focuses on permeable pavements [22] and land use [23]. Quantitative studies, on the other hand, typically rely on hydrodynamic process simulations to analyze the extent to which underlying surface factors influence waterlogging. Approaches such as system dynamics simulation [24] and machine learning [25] are also utilized to assess the factors affecting waterlogging. Although some studies have analyzed the factors that trigger waterlogging under various return periods, limited research has been conducted on the sensitivity ranges of underlying surfaces. Understanding the sensitivity range of the underlying surface is crucial for waterlogging management. Priority should be given to improving the highly sensitive underlying surface factors. Once the underlying surface factors have been improved to a certain extent and their sensitivity decreases, other factors can be considered for improvement. Focusing on this approach, targeted waterlogging control measures can be proposed, but also, considering their sensitivity, various combinations of underlying surface factors can be explored in order to achieve the best economic

benefits and ensure effective waterlogging control. Therefore, this study examines two case study areas, utilizes The United States Environmental Protection Agency Storm Water Management Model (EPA's SWMM, abbreviated: 'SWMM' in the following) for simulation, and employs the modified Morris method to analyze the sensitivity of land-use, drainage network capacity, and slope under different rainfall conditions. Simultaneously, the sensitivity ranges of these three factors are assessed to provide valuable insights for waterlogging prevention and control.

## 2. Materials and Method

### 2.1. Study Area

The study area is situated in Pingshan District, Shenzhen City, characterized by a tropical monsoon climate with substantial rainfall. The average annual rainfall measures 2073.5 mm, with the majority occurring from April to September, constituting 84% of the total precipitation. The average annual evaporation is 1345.7 mm, and the average annual temperature stands at 22.3 °C. Two communities were selected as the study areas. The Niujiaolong community (Figure 1a,b) spans a total area of 3.073 hectares. The terrain exhibits a higher elevation at the center and lower elevation around the periphery. Due to the presence of surrounding walls, the exchange of water with the external environment is disregarded. The land-use in the community is predominantly in the impervious area, accounting for 91.9%, followed by grassland (8.1%). The Zhuyuan community (Figure 1c,d) covers a total area of 23.4 hectares. The terrain features higher elevations in the north and south, with lower elevations in the middle. The eastern part of the community experiences lower elevations, ranging from 0.01 to 0.87 m lower than the surrounding areas, making it susceptible to water accumulation. Impervious area occupancy dominates the land-use in the community, accounting for 93.5%, followed by grassland (6%) and forest land (0.5%). As per the flood control and drainage plan of Pingshan District, the study area must adhere to the rainfall control standard with a return period of 5 years. The high level of urbanization has resulted in an increased proportion of impermeable surfaces, leading to diminished surface infiltration and the discharge of excess rainwater through drainage networks, thereby exacerbating waterlogging issues.

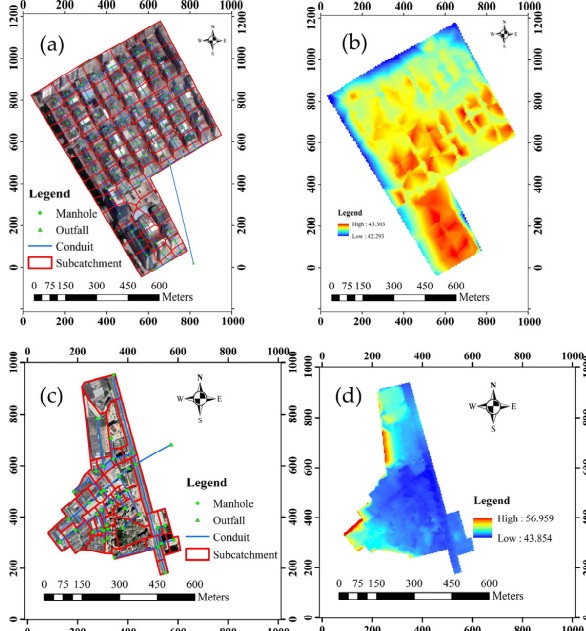

**Figure 1.** The geographic location of study area. (**a**) Underlying surface and drainage pipe networks of Niujiaolong community; (**b**) DEM of Niujiaolong community (m); (**c**) underlying surface and drainage pipe networks of Zhuyuan community; (**d**) DEM of Zhuyuan community (m).

### 2.2. Rainfall Data

The Chicago hyetograph method, initially proposed by Keifer in 1957 [26], is a designed storm method that determines the rainfall process for a specified intensity and duration based on a storm intensity formula and rainfall peak coefficient [27]. In China, the Chicago hyetograph method is widely employed in the planning and design of drainage systems and is recommended as a design rainfall method in the "Outdoor Drainage Design Standards" [28]. It has been extensively validated and is considered the most commonly used design rainfall method in China [29]. To explore the sensitive factors of waterlogging under various rainfall conditions, this study utilized the Chicago hyetograph method to provide rainfall input data. He, Wang [30] pointed out that over 70% of the rainfall in Shenzhen follows a single-peak pattern. Consequently, this study focused solely on designing single-peak rainfall patterns. By statistically analyzing 20 years (2001–2020) of minute-scale measured rainfall in Pingshan District, the comprehensive rainfall peak coefficient was calculated to be 0.512, basically falling within the widely accepted range of 0.3 to 0.5. The recommended design rainfall formula from the "Shenzhen Rainstorm Intensity Formula and Calculation Charts 2015 Edition" was employed (Equation (1)).

$$i = \frac{8.701(1 + 0.594\,lgP)}{(t + 11.13)^{0.555}} \tag{1}$$

where $i$ is rainfall intensity, mm/min; $P$ is return period, year; $t$ is time, min. Thirty design rainfall events for 6 return periods (1, 2, 3, 5, 10, and 20 years) under 5 rainfall durations (60, 90, 120, 150, and 180 min) are calculated. It is worth noting that previous studies have indicated a decreasing influence of land-use on runoff as the return period increases [20,31,32]. Therefore, the maximum return period was set to 20 years.

### 2.3. Model Construction

There are several drainage models available, such as the storm water management model (SWMM), Infoworks ICM, MIKE, etc. While commercial software such as Infoworks ICM and MIKE can simulate two-dimensional surface runoff processes, their adoption may be constrained by high costs. To address this concern, this study employs the open-source simulation SWMM mode. It has been extensively applied in various areas of urban water disaster [33], feasibility studies for low-impact development [34], design of detention ponds [35], and planning for drainage and flood prevention [36]. In this study, SWMM 5.1 was utilized to construct the drainage network models. For the Niujiaolong community, the generalized drainage network comprised 100 pipes, 88 nodes, and 1 outfall. The Thiessen polygon method was employed to delineate the sub-catchment, taking into account the uniform distribution of manholes. Regarding the Zhuyuan community, the generalized drainage network encompassed 40 pipes, 43 nodes, and 1 outfall. The initial delineation of sub-catchment was based on remote sensing imagery and further refined through field surveys. Parameter selection adhered to the standards for the design of outdoor wastewater engineering [28] and the research findings of Wei, Chun-xin [37]. The roughness values for grassland and impervious area were set at 0.2 and 0.012, respectively. Based on data obtained from geographical surveys of pipe network inspection wells, the roughness of the concrete pipes was determined as 0.013. Surface infiltration was calculated using the Horton method. The parameters used are presented in Table 1.

**Table 1.** The key parameters and value of the SWMM model.

| Sub-Catchment | | Horton Model | | Outfall | | Pipe | |
| --- | --- | --- | --- | --- | --- | --- | --- |
| Property | Value | Property | Value | Property | Value | Property | Value |
| N-Imperv | 0.012 | Max.Infil.Rate | 75 | Tide gate | NO | N | 0.013 |
| N-Perv | 0.2 | Min.Infil.Rate | 4 | Type | FREE | | |
| Dstore-Imperv | 2 | Decay Constant | 2 | | | | |
| Dstore-Perv | 7 | Drying Time | 7 | | | | |
| %Zero-Imperv | 25 | | | | | | |

*2.4. Sensitivity Analysis*

2.4.1. Sensitivity Analysis Method

Sensitivity analysis methods can be categorized into two types: global sensitivity analysis and local sensitivity analysis. Global sensitivity analysis methods, such as Sobol's method, the extended Fourier amplitude sensitivity test (EFAST), and the generalized likelihood uncertainty estimation (GLUE), are commonly utilized in hydrodynamic models [38–40]. These methods consider the coupling effects among parameters and provide a comprehensive understanding of the results. However, they require extensive computations. Conversely, local sensitivity analysis methods involve fewer computations, with the Morris screening method being the most commonly employed approach [41]. Morris screening method is a random one-factor-at-a-time design. Only one parameter $x_i$ is changed, and induced onto the model outcome $y = y(x_1, x_2, \ldots x_n)$, which can then be attributed to such a modification by means of an elementary effect $e_i$ defined by:

$$e_i = y^* - y\Delta_i \tag{2}$$

where $y^*$ is the new outcome, $y$ is the previous outcome, $\Delta_i$ is the variation in the parameter $i$.

Francos [42] demonstrated that the Morris screening method achieves relatively high accuracy when the independent variables vary at a fixed step size. Considering the stochastic nature of variable values in the Morris method, which can influence the results, this study employs the modified Morris method [43]. The modified Morris method assesses its sensitivity by analyzing the extent of variation in the dependent variable through an independent variable with a fixed step size. When conducting sensitivity analysis, begin by selecting a parameter $x_i$ from the study parameters and perturb it by a fixed percentage step size. By applying different perturbation values, corresponding simulated results, $y_i$ are obtained (refer to Equation (3)). The average value of the sensitivity coefficient serves as the criterion for parameter sensitivity. Due to its simplicity and the ease of visually assessing sensitivity, the Morris method is widely utilized in sensitivity analysis of rainfall–flood models.

$$SN = \frac{1}{n}\sum_{i=1}^{n-1} \frac{(Y_{i+1} - Y_i)/Y_0}{(P_{i+1} - P_i)/100} \tag{3}$$

where $Y_i$ and $Y_{i+1}$ are the output results of $i$ and $i+1$, respectively; $Y_0$ is the initial result of the parameters, $P_i$ and $P_{i+1}$ are the change percentages of the parameter value when the model of $i$ and $i+1$ run, respectively; and $N$ is the number of model operations.

However, sensitivity varies with changes in parameters, and sensitivity decreases when parameter values exceed a certain range. In practical applications, the focus lies on areas of higher sensitivity. Hence, this study conducts a further investigation into the variation in $S_i$, which is calculated as follows:

$$S_i = \frac{(Y_{i+1} - Y_i)/Y_0}{(P_{i+1} - P_i)/100} \tag{4}$$

Global sensitivity analysis methods are capable of considering the impact of multiple factors on model output simultaneously and provide a comprehensive understanding of the sensitivity levels of each factor [44]. As a result, they are particularly suitable for hydrological models that involve numerous parameters [45]. However, since this study focuses on only three factors, the modified Morris method is employed for sensitivity analysis. To ensure accurate calculations, overflow is considered to occur only when the maximum overflow volume exceeds 1 m$^3$. This approach avoids significant errors in the calculated Morris values when the overflow volume is minimal.

2.4.2. Sensitivity Analysis Indicators

Previous studies have shown that land-use, pipe capacity, and slope are recognized as the primary factors influencing waterlogging [16]. In the field of waterlogging management, the three underlying surface factors are commonly regarded as the principal aspects requiring improvement. Gaining a comprehensive understanding of the sensitivity of these three factors enables us to formulate targeted waterlogging control measures, as well as ascertain the required extent of improvement. The changes in land-use significantly influence surface infiltration and surface roughness. As impervious areas increase, both surface infiltration and ground roughness also increase, leading to reduced runoff and a more gradual runoff process. Often, when impervious areas reach a certain threshold, their proportionate influence on runoff diminishes. This phenomenon has been highlighted in the study conducted by Zhang, Cheng [46]. Therefore, in this research, land-use is represented by the proportion of impervious area (P-Imperv). The drainage capacity of pipelines plays a critical role in influencing pipe runoff. In situations where the drainage capacity of pipelines is insufficient, it has a significant impact on the runoff process and the volume of node overflow. Adequately increasing the drainage capacity can have a substantial effect on both the runoff process and the volume of node overflow. However, when the drainage capacity of pipelines is sufficient, the amount of rainfall becomes a more influential factor in the runoff process and node overflow. Increasing the drainage capacity may have a lesser impact on the runoff process and node overflow. To quantify the drainage capacity, this study introduces the pipe volume per hectare (PV-H), and the specific calculation formula is provided in Formula (5). As for slope, it primarily affects runoff processes in the surface, where steeper slopes lead to reduced runoff travel times, thereby influencing the runoff process and overflow volumes. Investigating the impact of slope on the runoff process and overflow volumes is a crucial aspect of this study. The slope is characterized by mean slope and slope standard deviation (sd slope). In total, four characteristics are considered for sensitivity analysis.

Land-Use

Impervious areas limit rainfall infiltration, leading to substantial surface runoff during the runoff process. The low roughness of impervious areas results in shorter runoff time, which, in turn, affects the runoff process and node overflow. Therefore, this study employs the P-Imperv as a representative parameter to assess the sensitivity of land-use to waterlogging.

Drainage Capacity

Areas with limited drainage capacity are susceptible to waterlogging and are primarily influenced by pipe diameter and pipe network density. In order to examine the effect of drainage capacity on waterlogging, this study combines these two factors into a single parameter referred to as PV-H. This parameter represents the pipe volume per unit area and serves as an indicator of the carrying and drainage capacity of the pipes within the study area. The calculation equation for PV-H is as follows:

$$s = \frac{1}{A_t} \sum_{i=1}^{n} A_i L_i \tag{5}$$

where $A_i$ is the area of pipe $i$, $m^2$; $L_i$ is the length of pipe $i$, m; $A_t$ is the study area, ha.

Slope

Slope plays a significant role in influencing the runoff process, as gentler slopes result in longer runoff times. In this study, both the mean slope and slope standard deviation are chosen to characterize the overall variation in slope and topographical changes within the study area. The calculations for these parameters are as follows:

$$S_m = \frac{1}{n}\sum_{i=1}^{n} S_i \tag{6}$$

$$S_{sd} = \sqrt{\frac{1}{n-1}\sum_{i=1}^{n}(S_{i-}S_m)^2} \tag{7}$$

where $S_m$ is the mean slope, $n$ is the total number of subcatchments, $S_i$ is the slope of subcatchment $i$, and $S_{sd}$ is the slope standard deviation.

Characteristics Parameters and Evaluation Criteria

The initial values of the actual parameters in the study area serve as the base values. A fixed step size of 10% is then applied to perturb these parameters. In the case of the P-Imperv, the perturbation range is set from −80% to 0% of the initial value, considering that it cannot exceed 100%. For the remaining parameters, the perturbation range is set from −80% to +80%, while keeping the other parameters constant. The ranges of values for the four parameters under study are presented in Table 2.

**Table 2.** Disturbance results and step size of characteristics parameters for SWMM model.

| Characteristics | Niujiaolong Community | | | Zhuyuan Community | | |
|---|---|---|---|---|---|---|
| | Range | Step | Base Value | Range | Step | Base Value |
| P-Imperv | 18.4~91.9% | −10% | 91.9% | 18.9~94.5% | −10% | 94.5% |
| PV-H | 22.8~227.6 | ±10% | 113.8 | 13.8~124.2 | ±10% | 68.9 |
| Mean slope | 0.04~0.38 | ±10% | 0.21 | 0.16~1.44 | ±10% | 0.8 |
| Sd slope | 0.04~0.36 | ±10% | 0.2 | 0.16~1.46 | ±10% | 0.81 |

This study adopts evaluation criteria from Lenhart, Eckhardt [47] for sensitivity indicators, which are divided into four categories. The specific evaluation criteria are shown in Table 3.

**Table 3.** Parameter sensitivity level classification.

| Class | Index | Sensitivity |
|---|---|---|
| I | $0.00 \leq |S_n/S_i| < 0.05$ | Small to negligible |
| II | $0.05 \leq |S_n/S_i| < 0.20$ | Medium |
| III | $0.20 \leq |S_n/S_i| < 1.00$ | High |
| IV | $|S_n/S_i| \geq 1.00$ | Very high |

## 3. Results

### 3.1. Sensitivity Analysis of the P-Imperv

3.1.1. Sensitivity Changes with Rainfall Characteristics for the P-Imperv

Land-use is represented by the P-Imperv, which significantly influences surface runoff and consequently impacts waterlogging and flooding. Given the focus of waterlogging management on overflow volume and maximum pipe flow, this study examines the sensitivity of these two indicators to various factors. It is necessary to clarify that before sensitivity analysis, the classification of sensitivity can be obtained from the legend, and for cases of the same sensitivity, they are determined by the size of the points. A high sensitivity means that small changes significantly affect both overflow volume and maximum

pipe flow. As depicted in Figure 2, for the Niujiaolong community, the sensitivity of the P-Imperv to total overflow volume and maximum pipe flow gradually diminishes with increasing return period and rainfall duration. This decline can be attributed to rainfall increases, which reduces the proportion of soil infiltration relative to the total rainfall and consequently lowers the sensitivity of the P-Imperv. The sensitivity of the P-Imperv to total overflow volume ranges from 0.58 to 1.11, signifying a high or very high sensitivity. When rainfall is relatively decreased, the sensitivity of the P-Imperv to overflow is classified as very high (class IV in Figure 2), aligning with the findings of Zhang, Cheng [46]. This suggests that changing the P-Imperv is highly effective in waterlogging management. Similarly, the sensitivity of the P-Imperv to maximum pipe flow demonstrates high sensitivity, ranging from 0.2 to 0.78, peaking when the rainfall duration is 60 min and the return period is 1 year. Similar to the total overflow volume, it exhibits a decreasing sensitivity trend with higher rainfall. Reconstructing factors with lower sensitivity pose challenges in achieving desired management outcomes in waterlogging control. Based on the aforementioned observations, it can be concluded that the P-Imperv exhibits a high or very high sensitivity to total overflow volume and maximum pipe flow. Decreasing the P-Imperv is an effective measure for reducing node overflow and peak pipe flow. In comparison to the Zhuyuan community (Figure S1), the sensitivity of the P-Imperv to total overflow volume is basically classified as very high, and to maximum pipe flow as high. This distinction may arise from the Zhuyuan community's high vulnerability to waterlogging, resulting in a greater sensitivity of the P-Imperv to total overflow volume and maximum pipe flow.

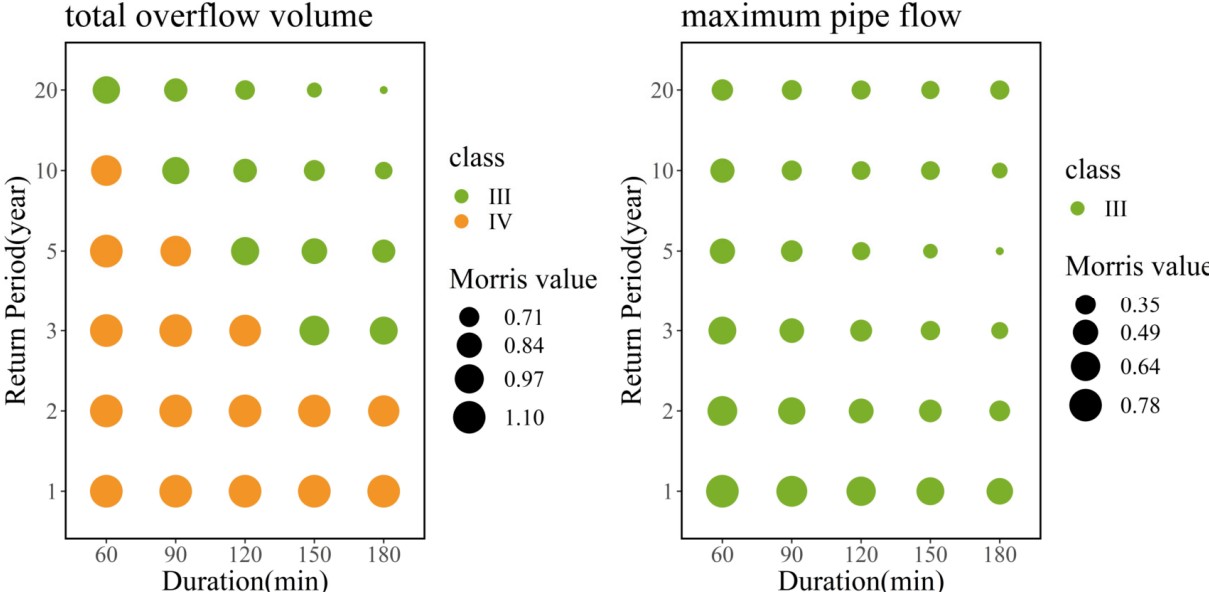

**Figure 2.** Sensitivity of the P-Imperv in Niujiaolong communities under different rainfall conditions. Due to the sensitivity classification criteria being based on absolute values, all figures in this study have been plotted after taking the absolute values.

### 3.1.2. Patterns of Change in Sensitivity with P-Imperv Variations

The sensitivity of changes in the P-Imperv to total overflow volume and maximum pipe flow was analyzed for multiple rainfall events (see Figure 3). Research findings indicate that when the rainfall return period exceeds a certain threshold, the influence of the P-Imperv on runoff significantly diminishes [48]. The present study demonstrates that the sensitivity of the P-Imperv to total overflow volume and maximum pipe flow exhibits instability for return periods of 10 and 20 years. This instability is likely attributed to the substantial rainfall, which renders the P-Imperv's influence negligible and results in unstable computational outcomes. Consequently, the sensitivity of the P-Imperv to total overflow volume and maximum pipe flow during 10-year and 20-year rainfall return

periods is not discussed under the analysis of factors changes in this study. The sensitivity of the P-Imperv to total overflow volume initially increases and then decreases with the increase in the P-Imperv. It exhibits a very high sensitivity when P-Imperv ranges from 36.8% to 82.7%. However, as the P-Imperv continues to increase, the sensitivity significantly decreases. Regarding the maximum pipe flow, as the P-Imperv increases, the sensitivity gradually decreases. At a relatively low P-Imperv, the sensitivity to maximum pipe flow is classified as high. Once the P-Imperv exceeds 82.7%, the sensitivity decreases to the medium. Similar to the total overflow volume, when the P-Imperv exceeds 82.7%, the increase in the P-Imperv has a significantly reduced impact on the sensitivity to the maximum pipe flow. Comparing the Zhuyuan community (Figure S2), the sensitivity of the P-Imperv to total overflow volume exhibits a similar increasing and then decreasing trend. Overall, a very high sensitivity is reached when the P-Imperv exceeds 56.7%. However, the sensitivity of the P-Imperv to maximum pipe flow shows a decreasing trend with the increase in the P-Imperv. It reaches a very high level when the P-Imperv is below 37.8%, and when the P-Imperv increases to 94.5%, the sensitivity decreases to medium. Both study areas exhibit similar patterns, but it is worth noting that a different P-Imperv is required to achieve a very high sensitivity. The difference can be attributed to the poorer pipe network conditions in the Zhuyuan community compared to the Niujiaolong community, making it more prone to node overflow. Consequently, this results in the observed discrepancy between the two areas. In regions liable to waterlogging, increasing the P-Imperv has a greater impact on node overflow.

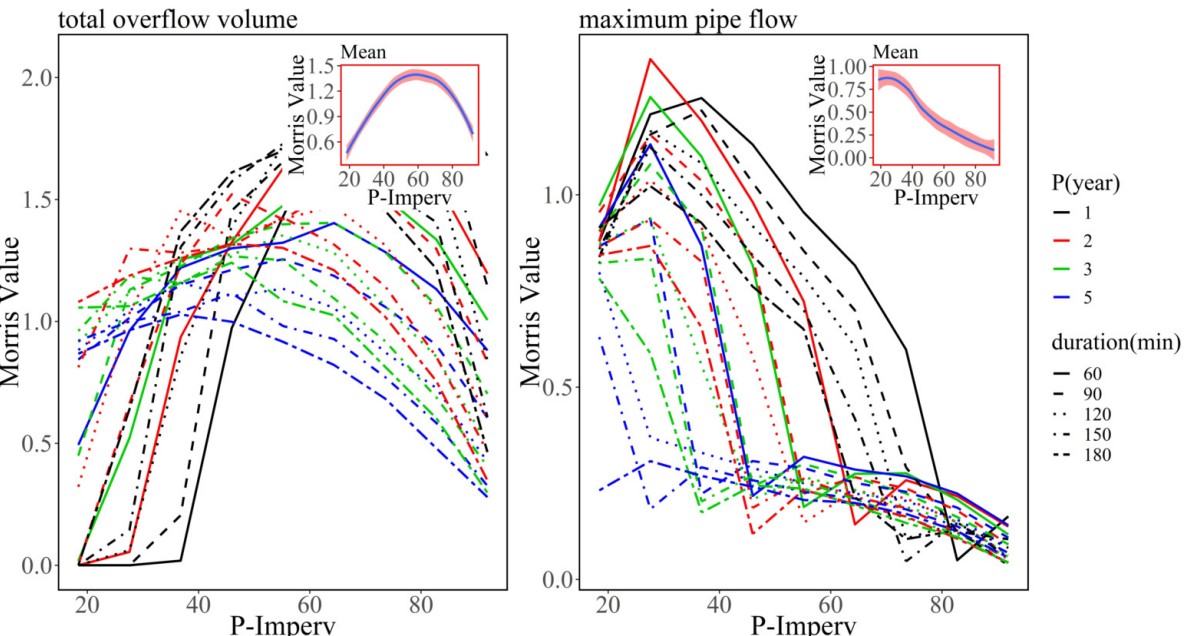

**Figure 3.** The patterns of change in sensitivity with the P-Imperv in Niujiaolong community.

*3.2. Sensitivity Analysis of PV-H*

3.2.1. Sensitivity Changes with Rainfall Characteristics for PV-H

The parameter PV-H is utilized in this study to represent the length and diameter of pipes in the study area. Figure 4 clearly demonstrates that for longer rainfall durations and smaller return periods, PV-H exerts a significant impact on total overflow volume. The sensitivity of PV-H to total overflow volume is overall classified as very high, highlighting that improving the pipe network is a key measure for controlling overflow. The highest sensitivity occurs when the rainfall duration is 180 min and the return period is 1 year, with a sensitivity value of 24.4. However, a different trend is observed for the maximum pipe flow. The sensitivity of PV-H to the maximum pipe flow ranges from 0.63 to 0.82, all falling within the high sensitivity classification. When the return period is small, the sensitivity

of PV-H to the maximum pipe flow is relatively low. As the return period increases, the sensitivity of PV-H to the maximum pipe flow also increases. This can be attributed to the fact that with a larger return period, inadequate pipe capacity leads to pipes operating at full capacity. Increasing the pipe diameter significantly enhances their drainage capacity, resulting in a higher sensitivity. Similar patterns are observed in the Zhuyuan community (Figure S3), with a very high sensitivity of PV-H to total overflow volume and a high sensitivity to the maximum pipe flow. Based on these results, it can be concluded that increasing PV-H is a crucial measure, whether to reduce node overflow volume or decrease the maximum pipe flow.

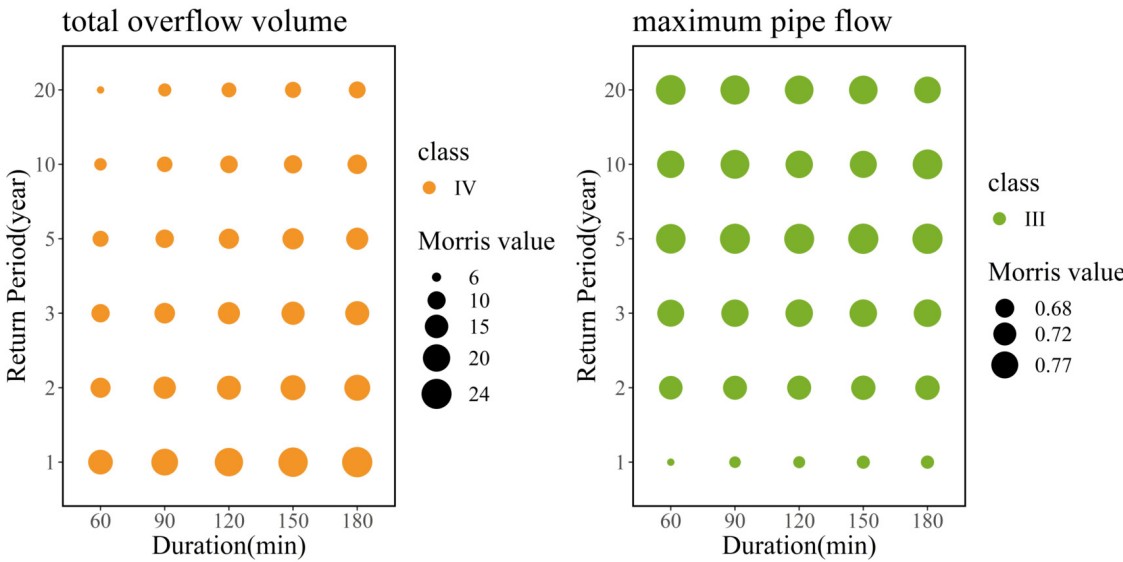

**Figure 4.** Sensitivity of the PV-H in Niujiaolong communities under different rainfall conditions.

### 3.2.2. Patterns of Change in Sensitivity with PV-H Variations

Figure 5 provides further insights into the impact of PV-H variations on sensitivity. It is observed that as PV-H increases, both the sensitivity to total overflow volume and the sensitivity to the maximum pipe flow exhibit a decreasing trend. Smaller PV-H values are associated with more severe node overflow, and increasing PV-H effectively reduces node overflow, resulting in a mean sensitivity value of 61.6, reaching up to very high sensitivity. However, as PV-H continues to increase, the sensitivity gradually decreases and reaches a high level when PV-H is 148. This is primarily because increased drainage capacity reduces the variation in node overflow. In some cases, node overflow even disappears during smaller rainfall events, leading to a decrease in sensitivity. The sensitivity of the maximum flow in the pipeline to the parameter PV-H exhibits a zigzag curve. It is speculated that a small PV-H can significantly enhance the flow in the pipeline, resulting in an upward trend in sensitivity. Nevertheless, as the drainage capacity of the pipeline approaches the rainfall intensity, the fluctuations in maximum flow become relatively smaller, leading to a decline in sensitivity. As PV-H continues to increase, the upstream water conveyance capacity improves, resulting in a substantial increase in the flow in the downstream pipeline. Consequently, the sensitivity exhibits a second increasing trend. However, when PV-H exceeds a certain threshold, its influence on the flow diminishes, resulting in a gradual decline in sensitivity. With sensitivity decreases, when PV-H reaches 79.7, sensitivity decreases to high. Similar patterns are observed in the Zhuyuan community (Figure S4), where the sensitivity of PV-H to total overflow volume reaches the high level when PV-H is increased to 89.6, and the sensitivity of PV-H to the maximum pipe flow reaches the high level when PV-H is increased to 20.7. Different study areas have distinct underlying surfaces, resulting in varying thresholds for sensitivity ranges. However, the general trend remains consistent: as PV-H increases, the sensitivity decreases.

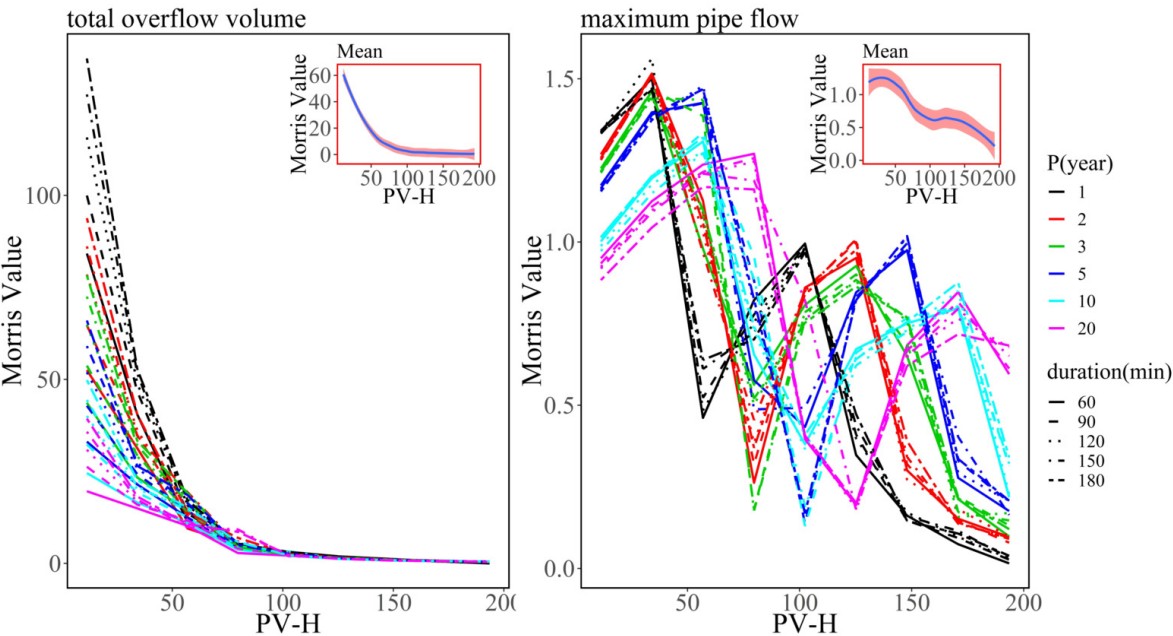

**Figure 5.** The patterns of change in sensitivity with the PV-H in Niujiaolong community.

### 3.3. Sensitivity Analysis of Slope

#### 3.3.1. Sensitivity Changes with Rainfall Characteristics for Slope

This study examines the sensitivity of waterlogging to ground slope by analyzing the mean slope and slope standard deviation of sub-catchments. Ground slope is a crucial factor influencing the runoff process [49], which subsequently impacts node overflow and pipe flow. Therefore, it is selected as a key indicator for analysis. As illustrated in Figure 6, the sensitivity of the mean slope to total overflow volume is generally classified as medium, gradually decreasing to small or negligible as rainfall increases. This decline in sensitivity can be attributed to the diminishing influence of slope on the runoff process under heavier rainfall, resulting in a reduced impact on waterlogging. However, the sensitivity of the mean slope to the maximum pipe flow is generally considered small or negligible. Thus, it can be concluded that ground slope has little effect on the maximum pipe flow. Given that the sensitivity of the slope standard deviation to both total overflow volume and maximum pipe flow is similar to that of the mean slope (see Figure S5), further discussion on slope standard deviation is omitted. Comparing these findings with those of the Zhuyuan community (see Figures S6 and S7), the sensitivity of slope to total overflow volume is classified as high, with sensitivity values ranging from 0.22 to 0.86. However, the sensitivity to the maximum pipe flow is classified as medium, with sensitivity values ranging from 0.05 to 0.17. Therefore, adjusting the ground slope may not be useful to prevent node overflow.

#### 3.3.2. Patterns of Change in Sensitivity with Slope Variations

In the presence of heavy rainfall, the influence of slope on runoff appears to be insignificant [50]. Therefore, this study does not explore the sensitivity of slope to total overflow and maximum pipe flow for a 10-year and 20-year rainfall return period under the analysis of factors changes, given the observed instability. The sensitivity to both total overflow volume and maximum pipe flow decreases as the mean slope increases, as depicted in Figure 7. When the mean slope reaches 0.07, the sensitivity to total overflow volume declines to a medium level, while the sensitivity to the maximum pipe flow is always below the medium. Similar patterns are observed for the slope standard deviation (see Figure S8). In comparison to the Zhuyuan community, when the mean slope increases to 1.05, the sensitivity of the mean slope to total overflow volume decreases to a medium level. Additionally, for relatively small mean slope values, the sensitivity can reach a very

high level (Figure S9), which is consistent with the findings for the slope standard deviation (Figure S10). These results indicate that in areas prone to waterlogging, appropriate measures can be implemented to mitigate the effects of the mean slope and ensure the attenuation of the runoff process.

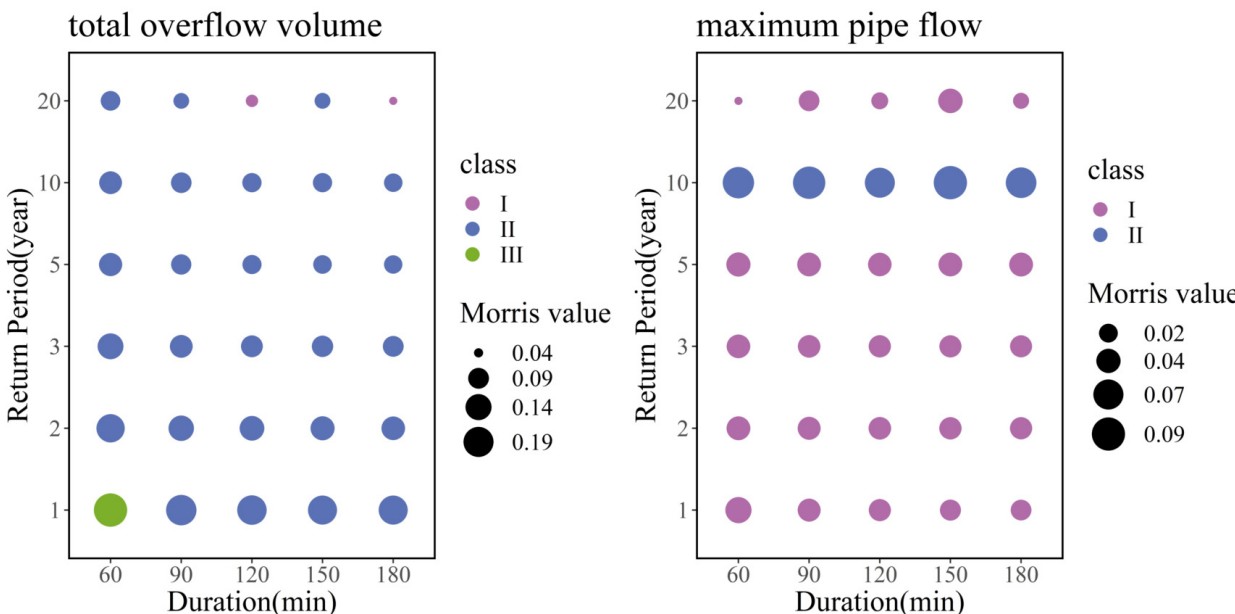

**Figure 6.** Sensitivity of the mean slope in Niujiaolong communities under different rainfall conditions.

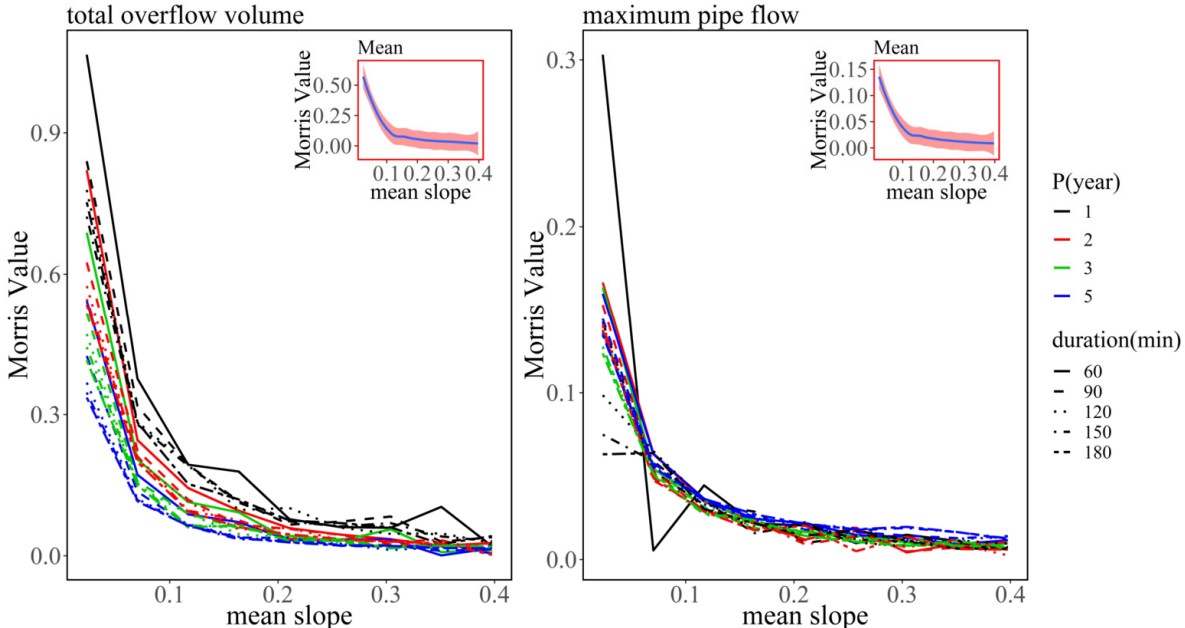

**Figure 7.** The patterns of change in sensitivity with the mean slope in Niujiaolong community.

## 4. Discussion

### 4.1. Selection of Key Factors for Urban Waterlogging Management

Surface modifications are crucial in mitigating urban waterlogging issues.

Understanding the sensitivity of surface factors to various rainfall conditions allows for tailored improvement strategies in different regions. Previous studies have highlighted inadequate drainage capacity and high impervious area as major contributors to waterlogging [17,23,24,51]. However, determining the most effective factor in different rainfall

conditions to address different causes remains unclear. In this study, the effects of the P-Imperv, PV-H, and slope on total overflow volume and maximum pipe flow were analyzed under different rainfall conditions. The findings indicate that pipe capacity is the primary factor requiring reconstruction. The sensitivity of PV-H to total overflow volume is consistently classified as very high, followed by the P-Imperv. For areas prone to waterlogging, the P-Imperv demonstrates a very high sensitivity to total overflow volume, making it a significant factor in waterlogging management. However, the impact of changing the impervious area may be limited in regions with intense rainfall, as the P-Imperv exhibits higher sensitivity under smaller rainfall events. On the other hand, the influence of slope on total overflow volume and maximum pipe flow is relatively low, making it less suitable as a factor for waterlogging management.

### 4.2. Sensitivity Thresholds of Factors

Understanding the sensitivity patterns of factors based on their parameter values is essential for analyzing the thresholds for modifying factors in waterlogging management. Traditionally, increasing drainage capacity is considered the primary measure for controlling waterlogging [52]. However, when the sensitivity reaches a certain level, alternative factors can be explored for reconstruction. For example, in the Zhuyuan community, when the PV-H is less than 117.1, increasing the pipe diameter or building new pipelines to improve the drainage capacity of the area should be considered. However, when the PV-H is higher than 117.1, its sensitivity to total overflow volume decreases to a middle level. If the P-Imperv exceeds 56.7% at this time, i reducing the P-Imperv to achieve control of the total overflow volume should be considered. Having a clear understanding of sensitivity patterns is valuable in identifying more suitable combinations of measures for effective waterlogging management while minimizing investment costs.

### 4.3. Prospects and Limitations of Research

With the increasing attention to urban waterlogging, the management and cost control of waterlogging remain constant concerns. Concepts such as LID and sponge cities can help control waterlogging to some extent by reducing surface runoff; however, they are often limited to smaller rainfall events. As rainfall intensity increases, the focus shifts to the capacity of drainage systems. When formulating waterlogging management policies, it is essential to fully utilize the advantages of various improvement measures and explore the feasibility of combining different strategies to reduce engineering investments and achieve effective waterlogging control. The study primarily analyzes the sensitivity of surface factors to total overflow volume and maximum pipe flow. Exploring the high-sensitivity intervals can determine the extent of waterlogging management, i.e., if the sensitivity decreases after reaching a certain level of reconstruction, it indicates the need to switch to alternative waterlogging management measures. However, this approach has limitations; it cannot solely analyze waterlogging management measures. It could serve as a constraint condition for multi-objective optimization algorithms when exploring combinations of waterlogging control strategies. Additionally, the method should be validated in different regions to assess its applicability.

### 5. Conclusions

Using the modified Morris method, this study analyzed the sensitivity of three main factors—P-Imperv, PV-H, and slope—that affect waterlogging under different rainfall conditions. Additionally, the study investigated sensitivity patterns based on parameter variations, providing valuable insights and methods for proposing comprehensive waterlogging management measures. The following noteworthy conclusions were drawn:

(1) PV-H demonstrates the highest sensitivity to total overflow volume and maximum pipe flow, followed by the P-Imperv, while mean slope and sd slope exhibit the lowest sensitivity. The sensitivity of PV-H to total overflow volume consistently remains very high and decreases with increasing rainfall, emphasizing its significance as a primary factor

in waterlogging management. Conversely, the sensitivity of PV-H to the maximum pipe flow increases with rainfall and maintains a high level. For the P-Imperv, the sensitivity to both total overflow volume and maximum pipe flow gradually decreases with increasing rainfall. In the Niujiaolong community, the P-Imperv exhibits a very high sensitivity to total overflow volume, while in the Zhuyuan community, due to its vulnerability to waterlogging, the sensitivity is higher compared to the Niujiaolong community. As for slope, both mean slope and sd slope yield similar results. In the Niujiaolong community, the sensitivity to total overflow volume is generally classified as medium, while the sensitivity to the maximum pipe flow is small to negligible. In the Zhuyuan community, the sensitivity to total overflow volume is basically high, while the sensitivity to the maximum pipe flow is medium. The order of sensitivity factors remains consistent in both study areas: PV-H, P-Imperv, and slope;

(2) Exploring the sensitivity patterns with respect to factor values reveals that the sensitivity of the P-Imperv to total overflow volume exhibits an increasing and then decreasing trend. When the P-Imperv ranges from 36.8% to 82.7% (Niujiaolong community) and from 82.7% to 94.5% (Zhuyuan community), the sensitivity is classified as very high. However, the sensitivity of the P-Imperv to the maximum pipe flow decreases with an increasing P-Imperv. Different study areas have different sensitivity ranges, and analyzing the threshold values of sensitivity ranges is beneficial for combining various waterlogging management measures. The sensitivity of PV-H to total overflow volume decreases with increasing PV-H. When PV-H decreases to 148 (Niujiaolong community) and 89.6 (Zhuyuan community), the sensitivity to total overflow volume decreases to a high level. However, the sensitivity of slope to both total overflow volume and maximum pipe flow remains lower than the very high level and should be considered to a lesser extent in waterlogging management.

**Supplementary Materials:** The following supporting information can be downloaded at: https://www.mdpi.com/article/10.3390/w15173131/s1, Figure S1: Sensitivity of the P-Imperv in Zhuyuan communities under different rainfall conditions; Figure S2: The patterns of change in sensitivity with the P-Imperv in Zhuyuan community; Figure S3: Sensitivity of the PV-H in Zhuyuan communities under different rainfall conditions; Figure S4: The patterns of change in sensitivity with the PV-H in Zhuyuan community; Figure S5: Sensitivity of the sd slope in Niujiaolong communities under different rainfall conditions; Figure S6: Sensitivity of the mean slope in Zhuyuan communities under different rainfall conditions; Figure S7: Sensitivity of the sd slope in Zhuyuan communities under different rainfall conditions; Figure S8: The patterns of change in sensitivity with the sd slope in Niujiaolong community; Figure S9: The patterns of change in sensitivity with the mean slope in Zhuyuan community; Figure S10: The patterns of change in sensitivity with the sd slope in Zhuyuan community.

**Author Contributions:** Y.L.: writing—original draft, model, data curation, and writing—review and editing. X.Q.: writing—original draft and methodology. Y.W.: model and data analysis. M.W.: writing—review and editing. All authors have read and agreed to the published version of the manuscript.

**Funding:** This research was funded by National Key R&D Plan of China, grant number 2021YFC3001400.

**Data Availability Statement:** Please contact the corresponding author for data.

**Acknowledgments:** We also thank the anonymous reviewers and the associated editor for providing insightful comments that helped to improve the manuscript.

**Conflicts of Interest:** The authors declare no conflict of interest.

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
