# Peer review of "Exploring the Sensitivity Range of Underlying Surface Factors for Waterlogging Control"

_water, doi:10.3390/w15173131_

Round 1

Reviewer 1 Report

The research paper on "Exploring the Sensitive Range of Underlying Surface Factors for Waterlogging Control" uses the modified Morris method to assess the sensitivity of key factors responsible for waterlogging. Following are key observations/ comments on the MS:

1. Give reference for modified Morris method and briefly illustrate its principle.

2. Explicitly define the significance of size of dots in figure 2 and 4.

3. Define the unit of elevation in figure 1.

4. Illustrate the reason for zig-zag curve pattern as demonstrated in figure 3, 5 and 7.

Several statements within MS needs restructuring e.g., 

Abstract: This study aims to analyze the sensitivity and sensitive range of management measures under different rainfall conditions, with the purpose of establishing a foundation for their combined implementation.

Introduction (line 29): From 2008 to 2018, waterlogging incidents transpired annually in over 150 cities across China. (Check usage of transpire).

Author Response

Detailed response to comments(ID: water-2538099)

Dear Reviewer 1:

Thank you for your comments concerning our manuscript entitled “Exploring the Sensitive Range of Underlying Surface Factors for Waterlogging Control” (ID: water-2538099). Your comment is valuable and very helpful for revising and improving our paper, as well as the important guiding significance to our researches. We have studied comments carefully and have made correction which we hope meet with approval.

Comments and Suggestions for Authors

The research paper on "Exploring the Sensitive Range of Underlying Surface Factors for Waterlogging Control" uses the modified Morris method to assess the sensitivity of key factors responsible for waterlogging. Following are key observations/ comments on the MS:

  1. Give reference for modified Morris method and briefly illustrate its principle.

Answer: Thank you for pointing it out. We added the reference for modified Morris method and gave a briefly illustration for it in lines 195 to 197.

Action: The modified Morris method assesses its sensitivity by analyzing the extent of variation in the dependent variable through an independent variable with a fixed step size.

  1. Explicitly define the significance of size of dots in figure 2 and 4.

Answer: Thank you for your advice. We added some contents to define the significance of size of dots in figure 2 and 4 in lines 285 to 289.

Action: It is necessary to clarify that before sensitivity analysis, the classification of sensitivity can be obtained from the legend, and for cases of the same sensitivity, they are deter-mined by the size of the points. A high sensitivity means that small changes will significantly affect both overflow volume and maximum pipe flow.

  1. Define the unit of elevation in figure 1.

Answer: Thank you for your advice. We added the unit of elevation in Figure 1 in the Figure Caption.

Action:

Figure 1. The geographic location of study area. (a) Underlying surface and drainage pipe networks of Niujiaolong community; (b) DEM of Niujiaolong community (m); (c) Underlying surface and drainage pipe networks of Zhuyuan community; (b) DEM of Zhuyuan community (m).

  1. Illustrate the reason for zig-zag curve pattern as demonstrated in figure 3, 5 and 7.

Answer: Thank you for your question. Explaining the zig zag curve is crucial for understanding the patterns of sensitivity. We explained it in the paper. Please refer to lines 318 to 326, 380 to 390 and 423 to 426 for details

Action:

Lines 318 to 326

Research findings indicate that when the rainfall return period exceeds a certain threshold, the influence of the P-Imperv on runoff significantly diminishes [1]. The present study funded that the sensitivity of the P-Imperv to total overflow volume and maximum pipe flow exhibits instability for return periods of 10 and 20 years. This instability is likely attributed to the substantial rainfall, which renders the P-Imperv‘s influence negligible and results in unstable computational outcomes. Consequently, the sensitivity of the P-Imperv to total overflow volume and maximum pipe flow during 10-year and 20-year rainfall return periods is not discussed in this study.

Lines 380 to 390

The sensitivity of the maximum flow in the pipeline to the parameter PV-H exhibits a zig-zag curve. It is speculated that a small PV-H can significantly enhance the flow in the pipeline, resulting in an upward trend in sensitivity. Nevertheless, as the drainage capacity of the pipeline approaches the rainfall intensity, the fluctuations in maximum flow become relatively smaller, leading to a decline in sensitivity. As PV-H continues to increase, the upstream water conveyance capacity improves, resulting in a substantial increase in the flow in the downstream pipeline. Consequently, the sensitivity exhibits a second increasing trend. However, when PV-H exceeds a certain threshold, its influence on the flow diminishes, resulting in a gradual decline in sensitivity. With sensitivity decreases, when PV-H reaches 79.7, sensitivity decreases to high.

Lines 423 to 426

In the presence of heavy rainfall, the influence of slope on runoff appears to be insignificant [2]. Therefore, this study does not explore the sensitivity of slope to total overflow and maximum pipe flow for a 10-year and 20-year rainfall return period under the analysis of factors changes, given the observed instability.

5.Comments on the Quality of English Language

Several statements within MS needs restructuring e.g.,

Abstract: This study aims to analyze the sensitivity and sensitive range of management measures under different rainfall conditions, with the purpose of establishing a foundation for their combined implementation.

Introduction (line 29): From 2008 to 2018, waterlogging incidents transpired annually in over 150 cities across China. (Check usage of transpire).

Answer: Thank you for pointing it out. We have checked the whole contents to ensure that the English writing is correct.

Action:

Thus, this study aims to analyze the sensitivity and sensitive range of management measures under different rainfall conditions, focusing on establishing a foundation for their combined implementation.

Moreover, frequent occurrences of heavy rainfall have further exacerbated the incidence of waterlogging events, e.g., from 2008 to 2018, a large amount of waterlogging incidents were reported annually in over 150 cities across China.

References

  1. Zhang, Q., et al., Slope Runoff Process and Regulation Threshold under the Dual Effects of Rainfall and Vegetation in Loess Hilly and Gully Region. Sustainability, 2023. 15(9): p. 7582.
  2. Zhao, Q., et al., Effects of rainfall intensity and slope gradient on erosion characteristics of the red soil slope. Stochastic Environmental Research and Risk Assessment, 2015. 29(2): p. 609-621.

Reviewer 2 Report

1.     The introduction lacks a clear statement of the specific research gap or objective of the study. Please guide the reader in understanding the specific contribution of this study to the existing body of knowledge.

2.     Please Explain why the selection of the three factors (land use, drainage network capacity, and slope) were chosen and how they relate to previous research on waterlogging sensitivity.

3.     Please simplify Figure 1. It is unnecessary to provide the two maps in the left. Show the coordinates instead.

4.     Please explain why SWMM is selected but not other models.

5.     Some figures are very messy, e.g., Figures 3, 5, 7. Please improve the quality.

6.     The paper lacks a clear discussion on the practical implications of these findings for waterlogging management measures. The authors should elaborate on how the identified sensitivity patterns can inform decision-making and help develop effective and targeted waterlogging management strategies.

7.     Please Explain the physical processes and mechanisms driving the sensitivity patterns of the factors (PV-H, P-Imperv, and slope). This would add depth to the analysis and enhance the readers' understanding of the results.

Improvements are needed for English writing.

Author Response

Detailed response to comments(ID: water-2538099)

Dear Reviewer 2:

Thank you for your comments concerning our manuscript entitled “Exploring the Sensitive Range of Underlying Surface Factors for Waterlogging Control” (ID: water-2538099). Your comment is valuable and very helpful for revising and improving our paper, as well as the important guiding significance to our researches. We have studied comments carefully and have made correction which we hope meet with approval.

1.The introduction lacks a clear statement of the specific research gap or objective of the study. Please guide the reader in understanding the specific contribution of this study to the existing body of knowledge.

Answer: Thank you for pointing it out. We have provided a detailed description of the research objective and specific contribution in the article in lines 94 to 101.

Action: Understanding the sensitive range of the underlying surface is crucial for waterlogging management. Priority should be given to improving the highly sensitive underlying surface factors. Once the underlying surface factors have been improved to a certain extent and their sensitivity decreases, other factors can be considered for improvement. Focusing on this approach, targeted waterlogging control measures can be proposed, but also, considering their sensitivity, various combinations of underlying surface factors can be explored in order to achieve the best economic benefits and ensure effective waterlogging control.

2.Please Explain why the selection of the three factors (land use, drainage network capacity, and slope) were chosen and how they relate to previous research on waterlogging sensitivity.

Answer: Thank you for your advice. Previous studies have rarely analyzed the sensitivity of underlying surface factors, and most of them focused on causes of waterlogging (we have explained in lines 82 to 94). So, in this paper, we added some contents to explain why we choose the three factors, and added the relationship between three factors and waterlogging sensitivity (lines 221 to 225).

Action: In the field of waterlogging management, the three underlying surface factors are commonly regarded as the principal aspects requiring improvement. Gaining a comprehensive understanding of the sensitivity of these three factors enables us to formulate targeted waterlogging control measures, as well as ascertain the required extent of improvement.

3.Please simplify Figure 1. It is unnecessary to provide the two maps in the left. Show the coordinates instead.

Answer: Thank you for your advice. We have redrawn Figure 1.

Action:

Figure 1. The geographic location of study area. (a) Underlying surface and drainage pipe networks of Niujiaolong community; (b) DEM of Niujiaolong community (m); (c) Underlying surface and drainage pipe networks of Zhuyuan community; (b) DEM of Zhuyuan community (m).

4.Please explain why SWMM is selected but not other models.

Answer: Thank you for pointing it out. We further explained several models that can be used for this study in the article, and briefly introduced the problems of other models, thereby explaining why we chose the SWMM model (line 157 to 161).

 Action: There are several drainage models available, such as the Storm Water Management Model (SWMM), Infoworks ICM, MIKE, etc. While commercial software like Infoworks ICM and MIKE can simulate two-dimensional surface runoff processes, their adoption may be constrained by high costs. To address this concern, this study employs the open-source simulation SWMM mode.

5.Some figures are very messy, e.g., Figures 3, 5, 7. Please improve the quality.

Answer: Thank you for your advice. The study examines the impact of the P-Imperv and slope on the total overflow volume and maximum pipe flow while elucidating the observed instability. As a result, the sensitivity results for the 10-year and 20-year return periods were omitted from Figures 3 and 7, aiming to improve figure quality and enable better visualization of the simulation outcomes. In contrast, Figure 5 features an analysis of the zig-zag curve patterns, all results in this particular figure were reserved.

Action:

Lines 318 to 326

Research findings indicate that when the rainfall return period exceeds a certain threshold, the influence of the P-Imperv on runoff significantly diminishes [1]. The present study funded that the sensitivity of the P-Imperv to total overflow volume and maximum pipe flow exhibits instability for return periods of 10 and 20 years. This instability is likely attributed to the substantial rainfall, which renders the P-Imperv‘s influence negligible and results in unstable computational outcomes. Consequently, the sensitivity of the P-Imperv to total overflow volume and maximum pipe flow during 10-year and 20-year rainfall return periods is not discussed in this study.

Lines 380 to 390

The sensitivity of the maximum flow in the pipeline to the parameter PV-H exhibits a zig-zag curve. It is speculated that a small PV-H can significantly enhance the flow in the pipeline, resulting in an upward trend in sensitivity. Nevertheless, as the drainage capacity of the pipeline approaches the rainfall intensity, the fluctuations in maximum flow become relatively smaller, leading to a decline in sensitivity. As PV-H continues to increase, the upstream water conveyance capacity improves, resulting in a substantial increase in the flow in the downstream pipeline. Consequently, the sensitivity exhibits a second increasing trend. However, when PV-H exceeds a certain threshold, its influence on the flow diminishes, resulting in a gradual decline in sensitivity. With sensitivity decreases, when PV-H reaches 79.7, sensitivity decreases to high.

Lines 423 to 426

In the presence of heavy rainfall, the influence of slope on runoff appears to be insignificant [2]. Therefore, this study does not explore the sensitivity of slope to total overflow and maximum pipe flow for a 10-year and 20-year rainfall return period under the analysis of factors changes, given the observed instability.

6.The paper lacks a clear discussion on the practical implications of these findings for waterlogging management measures. The authors should elaborate on how the identified sensitivity patterns can inform decision-making and help develop effective and targeted waterlogging management strategies.

Answer: Thank you for pointing it out. It is necessary to elaborate how the sensitivity patterns provide information for decision-making. Therefore, in the discussion, we use the Zhuyuan community as a case study to illustrate how to achieve a combination of waterlogging management methods to reduce node overflow while controlling costs. (lines 463 to 468)

Action:

Delete: For example, in the Zhuyuan community, when PV-H reaches 117.1, the sensitivity to total overflow volume decreases to medium level, indicating that reducing node overflow should be considered by focusing on P-Imperv.

Add: For example, in the Zhuyuan community, when the PV-H is less than 117.1, it can be considered to increase the pipe diameter or build new pipelines to improve the drainage capacity of the area. However, when the PV-H is higher than 117.1, its sensitivity to total overflow volume decreases to middle level. If the P-Imperv exceeds 56.7% at this time, it should be considered to reduce the P-Imperv to achieve control of the total overflow volume.

7.Please Explain the physical processes and mechanisms driving the sensitivity patterns of the factors (PV-H, P-Imperv, and slope). This would add depth to the analysis and enhance the readers' understanding of the results.

Answer: Thank you for pointing it out. We also think the physical processes and mechanisms driving the sensitivity patterns of the factors is important for our paper. Therefore, we further elaborated in the article (lines 225 to 244).

Action:

The changes in land use significantly influence surface infiltration and surface roughness. As impervious areas increase, both surface infiltration and ground roughness also increase, leading to reduced runoff and a more gradual runoff process. Often, when impervious areas reach a certain threshold, their proportionate influence on runoff diminishes. This phenomenon has been highlighted in the study conducted by Zhang, Cheng [3]. Therefore, in this research, land use is represented by the Proportion of Impervious Area (P-Imperv). The drainage capacity of pipelines plays a critical role in influencing pipe runoff. In situations where the drainage capacity of pipelines is insufficient, it has a significant impact on the runoff process and the volume of node overflows. Adequately increasing the drainage capacity can have a substantial effect on both the runoff process and the volume of node overflows. However, when the drainage capacity of pipelines is sufficient, the amount of rainfall becomes a more influential factor in the runoff process and node overflows. Increasing the drainage capacity may have a lesser impact on the runoff process and node overflows. To quantify the drainage capacity, this study introduces the Pipe Volume per Hectare (PV-H), and the specific calculation formula is provided in Formula 5. As for slope, it primarily affects runoff processes in the surface, where steeper slopes lead to reduced runoff travel times, thereby influencing the runoff process and overflow volumes. Investigating the impact of slope on the runoff process and overflow volumes is a crucial aspect of this study. The slope is characterized by mean slope and slope standard deviation (sd slope).

  1. Improvements are needed for English writing.

Answer: Thank you for pointing it out. We have checked the whole contents to ensure that the English writing is correct.

References

  1. Zhang, Q., et al., Slope Runoff Process and Regulation Threshold under the Dual Effects of Rainfall and Vegetation in Loess Hilly and Gully Region. Sustainability, 2023. 15(9): p. 7582.
  2. Zhao, Q., et al., Effects of rainfall intensity and slope gradient on erosion characteristics of the red soil slope. Stochastic Environmental Research and Risk Assessment, 2015. 29(2): p. 609-621.
  3. Zhang, H., et al., Effects of Impervious Surface on the Spatial Distribution of Urban Waterlogging Risk Spots at Multiple Scales in Guangzhou, South China. Sustainability, 2018. 10(5): p. 1589.

Reviewer 3 Report

Review of the manuscript entitled “Exploring the Sensitive Range of Underlying Surface Factors for Waterlogging Control”. This manuscript has the ambitious aim to analyze the sensitivity and sensitive range of various management measures under different rainfall scenarios using a modified Morris method. Overall, the manuscript is well-written, well-structured, and well-suited to the journal’s readership.

Some minor errors include spacing between reference numbers [5] etc which should be addressed.

Line 95:  Please write out the SWMM model's full name as “The United States Environmental Protection Agency Storm Water Management Model” at first use of the abbreviation. 

Line 96: Morris method should be capitalized “M” as this is a surname.

Line 106: No space between 22.3  & °C  should be “22.3°C”

Line 111: Please write the percent sign following the number without any space between i.e., 91.9%. Please check all the manuscript for this mistake.

Please tell us the current population data of both study areas. If at hand please indicate the population and urbanization growth rates also.

Line 142: delete ,,

Please provide a section that outlines the limitations of your study

Please provide some basic policy implications for both urban planning and waterlogging mitigation in terms of LID, sponge city concepts, etc.

Author Response

Detailed response to comments(ID: water-2538099)

Dear Reviewer 3:

Thank you for your comments concerning our manuscript entitled “Exploring the Sensitive Range of Underlying Surface Factors for Waterlogging Control” (ID: water-2538099). Your comment is valuable and very helpful for revising and improving our paper, as well as the important guiding significance to our researches. We have studied comments carefully and have made correction which we hope meet with approval.

Comments and Suggestions for Authors

Review of the manuscript entitled “Exploring the Sensitive Range of Underlying Surface Factors for Waterlogging Control”. This manuscript has the ambitious aim to analyze the sensitivity and sensitive range of various management measures under different rainfall scenarios using a modified Morris method. Overall, the manuscript is well-written, well-structured, and well-suited to the journal’s readership.

  1. Some minor errors include spacing between reference numbers [5] etc which should be addressed.

Answer: Thank you for pointing it out. We checked all the reference numbers and added the spacing.

  1. Line 95: Please write out the SWMM model's full name as “The United States Environmental Protection Agency Storm Water Management Model” at first use of the abbreviation.

Answer: Thank you for pointing it out. The full name of "SWMM" has been added in lines 102 to 104.

Action: Therefore, this study examines two case study areas, utilizes The United States Environmental Protection Agency Storm Water Management Model (EPA’s SWMM, abbreviated: ‘SWMM’ in the following) for simulation,

  1. Line 96: Morris method should be capitalized “M” as this is a surname.

Answer: Thank you for pointing it out. The “m” in 'Morris' is all replaced with capital letter.

  1. Line 106: No space between 22.3 °C should be “22.3°C”

Answer: Thank you for pointing it out. We have fixed it.

  1. Line 111: Please write the percent sign following the number without any space between i.e., 91.9%. Please check all the manuscript for this mistake.

Answer: Thank you for pointing it out. We have fixed it and checked all the content.

6.Please tell us the current population data of both study areas. If at hand please indicate the population and urbanization growth rates also.

Answer: Thank you for your advice. The population data for the study area is currently unavailable with precise figures. Instead, we have derived estimates by analyzing the number of residential buildings and floors using remote sensing imagery, assuming an average household size of 3.5 people. The Niujiaolong community comprises 63 buildings, each having 5 floors with one household per floor, yielding an estimated population of 1,103 people. Similarly, the Zhuyuan community comprises approximately 200 buildings, each with an average of 5 floors and one household per floor, resulting in an estimated population of about 3,500 people. Due to the inherent inaccuracies in the data, we have refrained from explicitly stating these figures in the paper.

7.Line 142: delete ,,

Answer: Thank you for pointing it out. We have fixed it.

  1. Please provide a section that outlines the limitations of your study

Answer: Thank you for your advice. We have added “4.3 Prospects and limitations of research” on the discussion, integrating Comment 8 and Comment 9 (as shown below) into a section ‑ Section 4.3 - to explains the limitations of the paper and some basic policy implications for both urban planning and waterlogging mitigation in terms of LID, sponge city concepts. (Lines 472 to 487)

Action:

With the increasing attention to urban waterlogging, the management and cost control of waterlogging remain constant concerns. Concepts like LID and sponge cities can help control waterlogging to some extent by reducing surface runoff; however, they are of-ten limited to smaller rainfall events. As rainfall intensity increases, the focus shifts to the capacity of drainage systems. When formulating waterlogging management policies, it is essential to fully utilize the advantages of various improvement measures and explore the feasibility of combining different strategies to reduce engineering investments and achieve effective waterlogging control. The study primarily analyzes the sensitivity of surface fac-tors to total overflow volume and maximum pipe flow. Exploring the high-sensitivity intervals can determine the extent of waterlogging management, i.e., if the sensitivity de-creases after reaching a certain level of reconstruction, it indicates the need to switch to alternative waterlogging management measures. However, this approach has limitations; it cannot solely analyze waterlogging management measures. It could serve as a constraint condition for multi-objective optimization algorithms when exploring combinations of waterlogging control strategies. Additionally, the method should be validated in different regions to assess its applicability.

9.Please provide some basic policy implications for both urban planning and waterlogging mitigation in terms of LID, sponge city concepts, etc.

Answer: Thank you for your advice. We highly endorse your advice. Some basic policy implications for waterlogging mitigation will sublimate this paper. (Lines 473 to 479)

Action:

Concepts like LID and sponge cities can help control waterlogging to some extent by reducing surface runoff; however, they are of-ten limited to smaller rainfall events. As rainfall intensity increases, the focus shifts to the capacity of drainage systems. When formulating waterlogging management policies, it is essential to fully utilize the advantages of various improvement measures and explore the feasibility of combining different strategies to reduce engineering investments and achieve effective waterlogging control.
